# The Characteristic of HBV Quasispecies Is Related to Occult HBV Infection of Infants Born to Highly Viremic Mothers

**DOI:** 10.3390/v16071104

**Published:** 2024-07-09

**Authors:** Yi Li, Yarong Song, Yiwei Xiao, Tong Wang, Lili Li, Minmin Liu, Jie Li, Jie Wang

**Affiliations:** 1Department of Microbiology & Infectious Disease Center, School of Basic Medical Sciences, Peking University Health Science Center, Beijing 100191, China; 2Department of Clinical Laboratory, Peking Union Medical College Hospital, Chinese Academy of Medical Sciences and Peking Union Medical College, Beijing 100730, China; 3Graduate School, Peking Union Medical College, Chinese Academy of Medical Science, Beijing 100730, China; 4NHC Key Laboratory of Medical Immunology, Peking University, Beijing 100191, China

**Keywords:** hepatitis B virus, mother-to-child transmission, occult hepatitis B virus infection, quasispecies, genetic distance

## Abstract

Although a combination of immunoprophylaxis and antiviral therapy can effectively prevent mother-to-child transmission (MTCT) of hepatitis B virus (HBV), a considerable number of infants born to highly viremic mothers still develop occult HBV infection (OBI). To uncover the virological factor and risk predictor for OBI in infants, we found that the diversity and complexity of maternal HBV quasispecies in the case group were lower than those in the control group. Mutations with significant differences between the two groups were most enriched in the NTCPbd and PreC regions. Genetic distance at the amino-acid level of the PreC region, especially the combination of three amino-acid mutations in the PreC region, could strongly predict the risk of OBI in infants. HBV quasispecies in OBI infants were highly complex, and the non-synonymous substitutions were mainly found in the RT and HBsAg regions. The sK47E (rtQ55R) and sP49L mutations in OBI infants might contribute to OBI through inhibiting the production of HBV DNA and HBsAg, respectively. This study found the potential virological factors and risk predictors for OBI in infants born to highly viremic mothers, which might be helpful for controlling OBI in infants.

## 1. Introduction

Currently, mother-to-child transmission (MTCT) remains one of the main routes of hepatitis B virus (HBV) infection. With the combination of immunoprophylaxis, including active and passive immunization, and antiviral therapy for highly viremic pregnant women, MTCT of HBV has been greatly controlled [1]. However, a considerable number of infants born to mothers with chronic HBV infection have occult HBV infection (OBI) [2,3,4,5,6,7]. In addition, it should be noted that there is no significant difference in the prevalence of OBI between infants whose mothers received tenofovir disoproxil fumarate antiviral therapy and those whose mothers did not receive antiviral therapy [8].

OBI is defined as the presence of replication-competent HBV DNA in the liver and/or serum of individuals who are hepatitis B surface antigen (HBsAg)-negative, using the currently available tests [9]. The discovery of infant OBI has raised new concerns about how long this state lasts and the exact long-term impact on young OBI individuals. Our recent study found that OBI was characterized by intermittent viremia in 236 infants born to HBsAg-positive mothers, and the incidence of OBI in these infants was 37.14%, 19.09%, 20.85%, 31.61%, 8.65%, and 12.71% at the ages of 7 months, 1, 2, 3, 4, and 8 years, respectively [10]. As we know, OBI not only presents a risk of HBV reactivation, but is also a potential source of HBV transmission [5]. More importantly, OBI is associated with an increased risk of chronic liver disease including hepatocellular carcinoma [9,11,12]. Therefore, understanding the risk factors of OBI in immunized infants born to mothers with chronic HBV infection will help control new HBV infections and reduce the burden of liver disease. Currently, there is no consensus on the risk factors for infant OBI. Our previous results showed that infants born to mothers with chronic HBV infection had a higher risk of OBI when their antibodies to HBsAg (anti-HBs) were lower than 100 mIU/mL at 7 months of age [7]. However, a considerable number of OBI infants still have high levels of anti-HBs; thus, it is not enough to explain the risk factors of infant OBI only with a low level of anti-HBs.

The low fidelity of HBV DNA polymerase during the reverse transcription of HBV pregenomic RNA leads to the existence of HBV quasispecies. It has been reported that the characteristics of HBV quasispecies can predict the seroconversion of hepatitis B e antigen (HBeAg) and the efficacy of antiviral therapy [13,14]. We previously showed that the diversity and complexity of HBV quasispecies in the “a” determinant region of HBV genome in the mothers of infants with immunoprophylaxis failure was significantly lower than those in the mothers of infants with immunoprophylaxis success [15]. However, the correlation between maternal HBV quasispecies characteristics and infant OBI remains unclear. In addition, because of the difficulty of amplifying the HBV genome in OBI infants whose viral load is extremely low, most studies on infant OBI only sequenced its *S* gene, thus leading to the characteristics of HBV quasispecies at the whole-genome level remaining unclear in OBI infants. In this context, understanding the characteristics of HBV quasispecies at the whole-genome level in OBI infants and their paired mothers may help us to understand the risk or virological factors behind the MTCT of OBI and the virological mechanism of intermittent viremia in OBI infants.

The level of maternal HBV DNA is associated with the incidence of OBI in infants [3]. To exclude the influence of HBV DNA levels on HBV quasispecies analyses, the present study enrolled infants born to HBeAg-positive and highly viremic (HBV DNA level > 6 log_10_IU/mL) mothers without antiviral treatment, and HBV DNA levels were comparable between the mothers of OBI infants and the mothers of the control group. The correlation between maternal HBV quasispecies characteristics and infant OBI was investigated using third-generation sequencing (TGS) technology. In addition, using HBV-specific DNA probes, we successfully sequenced the HBV genome in OBI infants using next-generation sequencing (NGS) technology and further compared the characteristics of HBV quasispecies between OBI infants and their mothers.

## 2. Materials and Methods

### 2.1. Study Population

As previously reported [7], 349 infants born to HBeAg-positive mothers with high HBV DNA levels (>6 log_10_IU/mL) were enrolled, and 323 of them had immunoprophylaxis success, which is usually defined as negative HBsAg and positive anti-HBs in infants aged 7–12 months. All enrolled pregnant women were negative for HAV, HCV, HDV, HEV, and HIV. Since our cohort was established before the update of the guideline which recommends antiviral therapy for highly viremic mothers [16], none of the mothers received antiviral therapy during pregnancy. All infants received three doses of 10 μg recombinant yeast-derived hepatitis B vaccine (Dalian Hissen Biopharm Inc., Dalian, China) at birth (within 12 h), 1 month, and 6 months, combined with one dose of 100 IU hepatitis B immunoglobulin (Hualan Biological Engineering Inc., Xinxiang, China). Among 323 infants with immunoprophylaxis success, the serum samples of 82 infants were available to measure HBV DNA levels at both 7 and 12 months of age, and 37.8% (31/82) of infants with positive HBV DNA at 7 and/or 12 months of age were considered OBI infants. Since most of the mothers in our cohort were infected with genotype C HBV, mothers infected with other genotypes of HBV were excluded to avoid the interference of heterogeneity among different genotypes. Finally, 15 OBI infants and their mothers were enrolled in the case group, and 23 infants without HBV infection and their mothers were enrolled in the control group, based on the serum samples available for TGS of the HBV genome. In addition, the serum samples of six infants with OBI were available for NGS of the HBV genome (Figure 1A).

This study conformed to the ethical guidelines of the 1975 Declaration of Helsinki and was approved by the Ethics Committee of Peking University Health Science Center (IRB00001052-12041 and IRB00001052-12042). Written informed consent was obtained from the parents of all infants.

### 2.2. HBV Serology, HBV DNA, and HBV RNA Tests

The levels of HBsAg, HBV DNA, and HBV RNA in the serum or the cell-based experiments were detected as described previously [17]. The primers used for measuring HBV DNA and 3.5 kb HBV RNA are listed in Appendix A.

### 2.3. HBV DNA Sequencing Analysis

For TGS, HBV DNA was amplified with the designed barcoded primer pairs. Next, the purified PCR products were quantified by Qubit 3.0 fluorometer, and then sent to Shanghai OE Biotech Co., Ltd. (OE Biotech, Shanghai, China) for library preparation and sequencing assays. For NGS, library preparation and sequencing assays were performed by MyGenostics Inc. (MyGenostics, Beijing, China), and the filtered data were mapped to the reference sequence of the genotype C HBV genome (AB014378) using the Burrows–Wheeler Aligner [18].

### 2.4. Quasispecies Characteristics Analysis

The HBV sequences were aligned by the Muscle program and divided into 11 coding and 7 noncoding regions (Appendix A). As reported in our previous study [15], the characteristics of HBV quasispecies were evaluated by complexity (Shannon entropy) and diversity (genetic distance). Quasispecies characteristics and complexity of single position were calculated by quasispecies analysis package [19].

### 2.5. Quasispecies Population and Mutation Analyses

Dominant and sub-dominant sequences were selected through the sequence frequency of each sample calculated by the PickRobustOTU model of quasispecies analysis package. The SNPs of HBV sequences in infants were detected by lofreq [20].

### 2.6. Plasmids

Plasmids carrying the sK47E (pBB4.5-HBV1.3-sK47E) and sP49L (pBB4.5-HBV1.3-sP49L) mutations were constructed by a site-directed mutation method using pBB4.5-HBV1.3 containing a wild 1.3-fold-length genome of genotype C HBV. The primers used for plasmid construction are listed in Appendix A.

### 2.7. Cell Culture, Transfection, and Western Blot

Cell culture, transfection, and Western blot were also performed, as we previously reported [17]. HBsAg was detected by horse anti-HBs (Abcam, Cambridge, MA, USA) or mouse anti-preS1 (Fitzgerald, Acton, MA, USA).

### 2.8. Statistical Analysis

Categorical variables were expressed as % (m/n) and examined by χ2/Fisher’s exact test. Non-normally distributed data were expressed as median and interquartile intervals and tested by Mann–Whitney *U* test. Data for mother–infant pairs were compared by Wilcoxon signed-rank test. All *p* values were two-tailed, and *p* < 0.05 was considered statistically significant. Receiver operating characteristic (ROC) curve analyses were performed by R (version 4.2.1) (https://cran.r-project.org, accessed on 1 May 2024).

## 3. Results

### 3.1. No Significant Difference in the Consensus Sequence of Full-Length HBV Genome Is Found between the Mothers of the Two Groups

In this prospective nested case-control study, 15 infants with OBI and their mothers were enrolled in the case group, and 23 infants without HBV infection and their mothers were enrolled in the control group, based on the serum samples available for TGS of the HBV genome (Figure 1A). The HBV serologic results for each sample are provided in Appendix A. There were no significant differences in the baseline characteristics and sequence numbers between the two groups (Table 1). Since genotypes B and C are the two major HBV genotypes in China, the phylogenetic tree analysis was performed using the reference sequences of genotype B (GenBank accession no. AB554017) and genotype C HBV (AB014378). The results revealed that all consensus sequences of the maternal HBV genome clustered with the reference sequence of genotype C HBV, and there was no clustering sign of HBV genome sequences between the two groups, suggesting that the HBV genome sequences between the two groups were similar (Figure 1B).

### 3.2. The Diversity and Complexity of HBV Quasispecies in the Mothers of OBI Infants Are Lower than Those of the Control Group

The characteristics of HBV quasispecies are usually evaluated at the nucleotide and amino-acid levels of diversity (genetic distance) and complexity (Shannon entropy) [15]. As shown in Appendix A, the genetic distances at the nucleotide level (ntd) of the full-length HBV genome and many functional regions, including presurface 1 (PreS1), sodium taurocholate cotransporting polypeptide (NTCP) binding domain (NTCPbd) of large-HBsAg, small-HBsAg (HBsAg), major hydrophilic region of HBsAg (MHR), reverse transcriptase domain (RT) of HBV DNA polymerase, core protein (HBc), and surface promoter II (SPII), were significantly lower in the case group than those in the control group (Figure 2A). Further, we found that the synonymous substitution rates (*dS*) of the NTCPbd and RT regions in the case group were significantly lower than those in the control group (Figure 2B). The non-synonymous substitution rates (*dN*) of the NTCPbd, HBsAg, MHR, RT, and HBc regions in the case group were significantly lower than those in the control group, and a downward trend in the PreS1 region was also found in the case group (Figure 2C). The genetic distances at the amino-acid level (aad) of the PreS2, NTCPbd, X protein (HBx), MHR, Precore (PreC), and HBc regions in the case group were also significantly lower than those in the control group (Figure 2D).

For the complexity of HBV quasispecies, the Shannon entropy at the nucleotide level (nts) of the NTCPbd and SPII regions in the case group was significantly lower than that in the control group, and there was a downward trend for the nts of full-length HBV genome, PreS1, and HBc regions in the case group (Figure 2E). Meanwhile, there was also a downward trend for Shannon entropy at the amino-acid level (aas) of the PreS1 and NTCPbd regions in the case group, even though there was no significant difference between the two groups (Figure 2F).

### 3.3. The NTCPbd and PreC Regions in the Mothers of OBI Infants Are More Conservative than Those of the Control Group

To investigate the role of single-position complexity of the maternal HBV genome in the occurrence of OBI in infants, the complexity of a single position was calculated for each mother and compared between the two groups. There were 130 positions with significant differences between the two groups (Figure 3A). Compared with the control group, the complexity of the 93 positions was significantly lower in the case group. These positions were mostly enriched in NTCPbd and PreC regions. Since NTCPbd and SPII regions overlapped and were included in the PreS1 region, the distributions of these positions in the PreS1, NTCPbd, and SPII regions were 37.63% (35/93), 26.88% (25/93), and 24.73% (23/93), respectively. As shown in Figure 3B, 36 mutations were distributed across 25 significantly different positions in the NTCPbd region. Among these mutations, 25 were amino-acid substitutions, including 10 non-synonymous substitutions and 15 deletion mutations, and 11 were synonymous substitutions. Only 5.56% (2/36) of the mutations were present in the case group and were synonymous substitutions. It is known that only amino-acid substitutions may affect the function of large-HBsAg, especially the affinity between PreS1 and its receptor NTCP. We found that 43.48% (10/23) of the mothers in the control group had amino-acid substitutions, including non-synonymous substitutions and deletion mutations.

Only the aad of the PreC region in the case group was significantly lower than that in the control group; however, there was no significant difference in the *dN* of the PreC region between the two groups (Figure 2C,D), indicating that deletion or insertion mutations might lead to a different aad of the PreC region between the two groups by causing a frameshift mutation or premature translation termination. Based on this, amino-acid sequences with significantly different proportions between the two groups were explored, and three amino-acid sequences were found: Seq_type 1, Seq_type 2, and Seq_type 3 (Figure 3C). Seq_type 1 and Seq_type 2 were sequences with premature translation termination caused by deletion mutations at positions nt1819–1822 and nt1817, respectively. Seq_type 3 was a sequence with deletion mutation in the entire PreC region. Therefore, Seq_types 1–3 did not express HBeAg. The proportion of these amino-acid sequences in the case group was significantly lower than that in the control group (Figure 3D).

The values of quasispecies characteristics and the proportion of PreC seq_type 1–3 that were significantly different between the two groups in predicting the risk of infant OBI were explored by a ROC curve. The results showed that quasispecies characteristics that were significantly different between the two groups could predict the risk of infant OBI (Figure 3E and Appendix A and Appendix A). The predictive value for the *dN* of the PreC region was the highest among these quasispecies characteristics, with an area under the ROC curve (AUC) of 0.881 (95%CI: 0.769–0.993, cut-off value: 0.1056). The proportion of PreC seq_types 1–3 also had a high predictive value for the risk of infant OBI. Moreover, the combination of seq_type 1–3 proportions showed the highest predictive value for the risk of infant OBI, with an AUC of 1 (cut-off value: 2.049%) (Figure 3E and Appendix A).

### 3.4. HBV Quasispecies in OBI Infants Are Highly Complex

So far, the characteristics of HBV quasispecies at the whole-HBV-genome level have not been reported in OBI infants because it is difficult to obtain the sequence of the whole HBV genome in OBI infants with low HBV DNA levels through traditional PCR-based sequencing methods. In this study, amplified HBV DNA was captured by biotinylated single-strand DNA probes, enriched by streptavidin-coated magnetic beads, and then sequenced by NGS programs to analyze the characteristics of HBV quasispecies in OBI infants.

Since there were six infants with sufficient serum samples for NGS analyses, the single-position complexities of the HBV genome in six infants were compared with those of their mothers. The formula used to calculate the complexity of a single position was normalized based on sequencing depth; thus, the sequencing results between infants and their mothers were comparable. As shown in Figure 4A, there were 912 positions with significant differences between the two groups. Among these positions, the complexity of 543 positions in mothers was significantly higher than that in infants, and the complexity of 369 positions in infants was significantly higher than that in their mothers. Although more positions with decreased complexity were found in infants, the absolute complexity of positions with increased complexity in infants was obviously higher than that in their mothers, indicating that the HBV genome in infants with OBI was unstable.

For positions with decreased complexity in infants, most of them were distributed in the HBc, PreS1, HBx, SPII, core promoter, basic core promoter, enhancer (EN) II, and PreC regions, while for positions with increased complexity in infants, most of them were distributed in the RT, ENI, HBsAg, X promoter, MHR, NTCPbd, and SPI regions (Figure 4B). It has been reported that the higher the complexity of HBV quasispecies, the weaker is the overall replication ability of HBV population [14,15]. Therefore, the increased complexities of RT and ENI regions were consistent with the low level of HBV DNA in OBI infants, and the increased complexities of HBsAg, MHR, and SPI regions were consistent with the characteristic of undetectable HBsAg in OBI infants. A total of 473 mutations were detected at 369 positions, with increased complexity in infants. Among these mutations, 376 were present in all six infants and were mainly distributed in the RT, ENI, and HBsAg regions (Figure 4C), and 119 of them caused amino-acid substitutions (Appendix A). As shown in Figure 4D, 25 amino-acid substitutions in the HBsAg region were detected only in infants with OBI. Three of these mutations were located in the MHR region, including the most well-known immune-escape mutant G145R, which is deficient in HBsAg secretion, antigenicity, and anti-HBs binding [21]; 56% (14/25) of these mutations were located in the transmembrane domain of HBsAg, which is related to HBsAg secretion [22]; and 32% (8/25) of them were located in the first cytosolic loop of HBsAg, which is related to HBsAg assembly [23]. Since the HBsAg region overlaps with the RT region, these mutations also caused 10 amino-acid substitutions in the RT region, such as sT5A (rtN13S) and sK47E (rtQ55R).

### 3.5. The Effects of HBsAg and RT Mutations in HBV Replication

Among the amino-acid substitutions in the HBsAg region present only in OBI infants, HBV sT5A (rtN13S), sP49L, and sK47E (rtQ55R) mutations were the top three mutations ranked by the median mutation frequency (Figure 4D). Since the sT5A (rtN13S) mutation did not affect HBV replication [22], the effects of sP49L and sK47E (rtQ55R) mutations on HBV replication were detected to explore the possible virological factors for infant OBI. The results showed that the sP49L mutation significantly reduced the levels of supernatant and intracellular HBsAg, whereas the sK47E (rtQ55R) mutation had no effect on HBsAg levels (Figure 5A,B). For HBV DNA, the sP49L mutation significantly decreased the level of supernatant HBV DNA but increased the level of intracellular HBV DNA, whereas the sK47E (rtQ55R) mutation decreased the levels of supernatant and intracellular HBV DNA (Figure 5C,D). Consequently, the sP49L mutation did not affect the level of total HBV DNA, including the supernatant and intracellular HBV DNA, whereas the sK47E (rtQ55R) mutation significantly reduced the level of total HBV DNA (Figure 5E). Both sP49L and sK47E (rtQ55R) mutations did not affect the level of intracellular 3.5 kb HBV RNA (Figure 5F).

These results indicated that the sP49L mutation might decrease the release of supernatant HBsAg and HBV DNA by reducing HBsAg expression, and that the sK47E (rtQ55R) mutation might decrease the level of total HBV DNA by affecting the reverse transcription of 3.5 kb pgRNA.

## 4. Discussion

With the development of sequencing technologies, TGS has been used to detect long DNA sequences. Compared to the clone-based sequencing method, which was previously considered the gold standard for virus quasispecies analyses, TGS can significantly improve sequencing depth [24]. Therefore, the performance of TGS in depicting the characteristics of HBV quasispecies is undoubtedly much better than that of the clone-based sequencing method. In the present study, TGS was used to analyze the sequence of the HBV genome in mothers with chronic HBV infection, and NGS was used to analyze the sequence of the HBV genome in OBI infants due to the low HBV DNA level.

Many infants born to mothers with chronic HBV infection have OBI despite immunoprophylaxis success [2,3,4,5,6,7]. The discovery of infant OBI has raised new concerns about how long this state lasts and the outcome for OBI individuals. Our previous studies showed that OBI in children was characterized by intermittent viremia [10,25]. Individuals with OBI may experience HBV reactivation in an immunosuppressed state and even develop cirrhosis and hepatocellular carcinoma under the procancer mechanism of OBI [9,11,12,26]. Therefore, OBI may pose a significant threat to the prevention and management of HBV infection, which needs to be controlled.

In this study, we found that the diversity and complexity of HBV quasispecies in the mothers of OBI infants were lower than those in the control group. Specifically, the diversities of the full-length HBV genome and regions including PreS1, NTCPbd, HBsAg, MHR, RT, HBc, and SPII in the mothers of OBI infants were significantly lower than those in the control group. The complexities of the NTCPbd and SPII regions in the mothers of OBI infants were significantly lower than those in the control group. MTCT is considered as a bottleneck event of HBV proliferation, and only a small proportion of HBV quasispecies can be transmitted to newborns [15,27]. The reduction in viral fitness always follows a bottleneck event, and can be compensated for by the subsequent enlargement of viral population [28]. However, the effect of Muller’s ratchet must be confronted, that is, a diverse and complex viral population at the initial time of infection will hardly maintain the dominant phenotype of quasispecies; thus, it will enter into the error catastrophe and eventually become extinct [29,30]. Therefore, HBV quasispecies with relatively low diversity and complexity are more likely to successfully establish an infection in the new host. In addition, the anti-HBs titers at 7 and 12 months of age were comparable between the case and control infants, although the diversity and complexity of maternal HBV quasispecies in the case group were lower than those in the control group. Therefore, the specific immune response might not be influenced by HBV quasispecies.

After comparing the single-position complexity of the HBV genome between the two groups of mothers, we found that the complexity of 93 positions in the mothers of OBI infants was significantly lower than that in the control group, and these positions were mostly enriched in the NTCPbd and PreC regions. NTCP is a functional receptor of HBV that enters hepatocytes; thus, its conservation is crucial for HBV to establish infection. The PreC region contains a Toll/IL-1 receptor-like domain that enables HBeAg to antagonize the Toll-like receptor signaling cascade, thereby allowing HBV to evade the innate immune response [31]. Although these mutations in the NTCPbd and PreC regions only account for a small portion of the virus population, bottleneck events will magnify their impact on the establishment of HBV infection. Therefore, the conservative NTCPbd and PreC regions of the maternal HBV quasispecies may contribute to the establishment of OBI in infants. Further, we found that the characteristics of HBV quasispecies and proportions of PreC seq_types could predict the risk of infant OBI. The combination of PreC seq_type 1–3 proportions showed the highest predictive value, with an AUC of 1 (cut-off value: 2.049%). However, considering the simplicity of clinical application, the aad of the PreC region was an optimal predictor for the risk of infant OBI, with an AUC of 0.881 (95%CI: 0.769–0.993, cut-off value: 0.1056).

Compared with their mothers, the number of positions with high complexity in OBI infants decreased obviously, and most were located in the HBc region. This phenomenon was consistent with the results of our previous study on the MTCT of overt HBV infection, in which the complexity of HBV quasispecies and the immune-escape mutations of the HBc region in immunoprophylaxis-failure infants was lower than that in their mothers [32]. For positions with high complexity in OBI infants, the absolute value of complexity was obviously higher than that in their mothers, and these positions were mainly located in the RT, ENI, and HBsAg regions. This indicated that the selective advantages of these mutations in infants might contribute to OBI; for example, the increased complexities of RT and ENI regions might lead to a low level of HBV DNA in OBI infants, and the increased complexity of the HBsAg region might lead to undetectable HBsAg in OBI infants. In this study, we first reported the existence of OBI-related mutations sP49L and sK47E (rtQ55R) in infants with OBI, except for the classical sG145R mutation. The sP49L mutation reduced HBsAg expression and the production of infectious HBV virions but did not affect HBV replication, which led to the retention of intracellular HBV DNA. The sK47E (rtQ55R) mutation reduced HBV DNA levels by inhibiting the reverse transcription of 3.5 kb pgRNA. Since both sP49L and sK47E (rtQ55R) mutations are located in a T-cell epitope region (aa28–51) of HBsAg [33], their emergence might be the result of the evolution of robust HBV quasispecies under T-cell immune pressure. However, this study was conducted in a relatively small number of OBI infants and their mothers, and given this limitation, the results will be more promising when further validated with a larger sample size of OBI infants and their mothers.

In summary, our findings provide the potential virological factors and risk predictors for OBI in infants born to highly viremic mothers. Moreover, this is the first study to provide the whole-genome landscape of HBV quasispecies in infants with OBI. Therefore, this study may help control OBI in infants born to highly viremic mothers and promote the achievement of the goal of eliminating viral hepatitis by 2030.

## Figures and Tables

**Figure 1 viruses-16-01104-f001:**
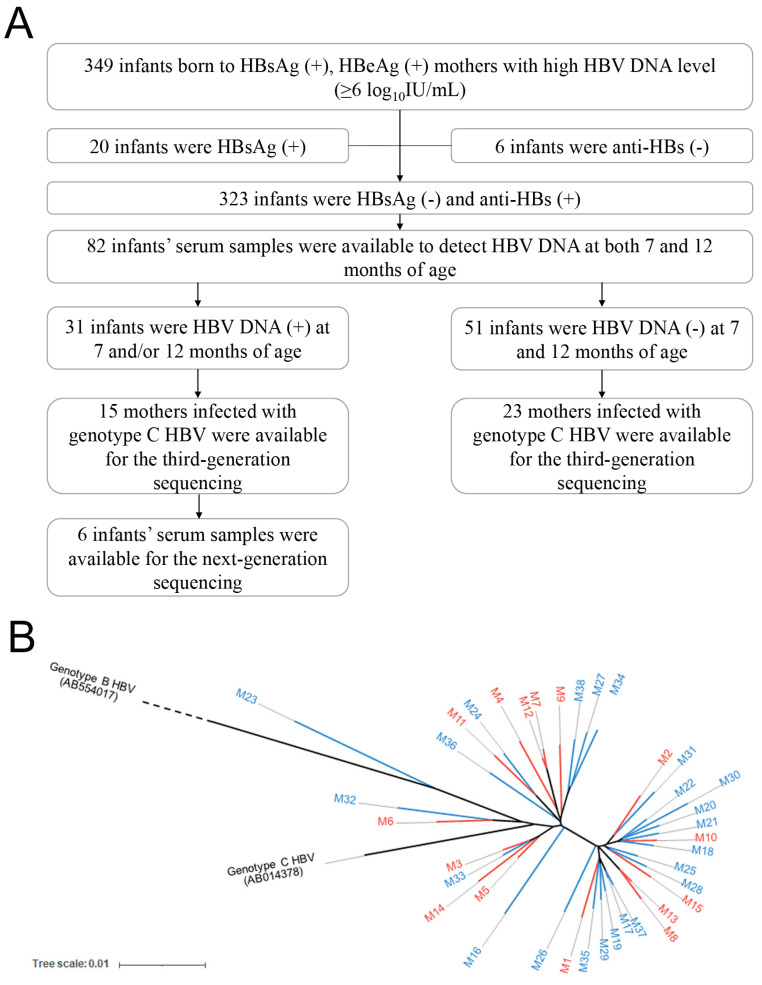
The enrollment flowchart of the cohort and the maximum likelihood phylogenetic tree of the HBV genome consensus sequences. (**A**) The enrolment flowchart of the cohort. (**B**) The maximum likelihood phylogenetic tree of the HBV genome consensus sequences in the two groups of mothers. The red and blue branches represent mothers in the case and control group, respectively.

**Figure 2 viruses-16-01104-f002:**
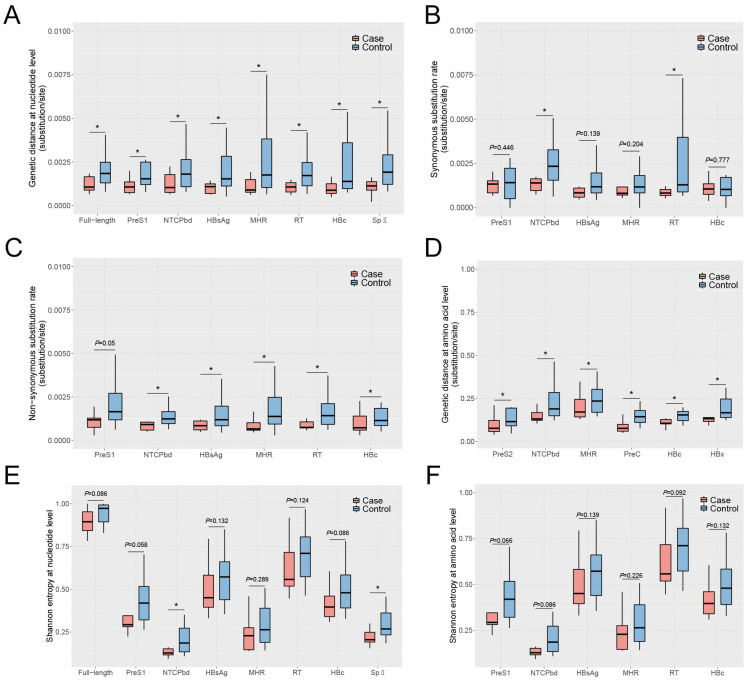
The characteristics of HBV quasispecies between the two groups. (**A**) The genetic distance at nucleotide level of full-length HBV genome, PreS1, NTCPbd, HBsAg, MHR, RT, HBc, and SPII regions between the two groups. (**B**). The synonymous substitution rates of PreS1, NTCPbd, HBsAg, MHR, RT, and HBc regions. (**C**) The non-synonymous substitution rates of PreS1, NTCPbd, HBsAg, MHR, RT, and HBc regions. (**D**) The genetic distance at amino-acid level of PreS2, NTCPbd, MHR, PreC, HBc, and HBx regions. (**E**) The Shannon entropy at nucleotide level of full-length HBV genome, PreS1, NTCPbd, HBsAg, MHR, RT, HBc, and SPII regions. (**F**) The Shannon entropy at amino-acid level of PreS1, NTCPbd, HBsAg, MHR, RT, and HBc regions. * Represents *p* < 0.05, Mann–Whitney *U* test.

**Figure 3 viruses-16-01104-f003:**
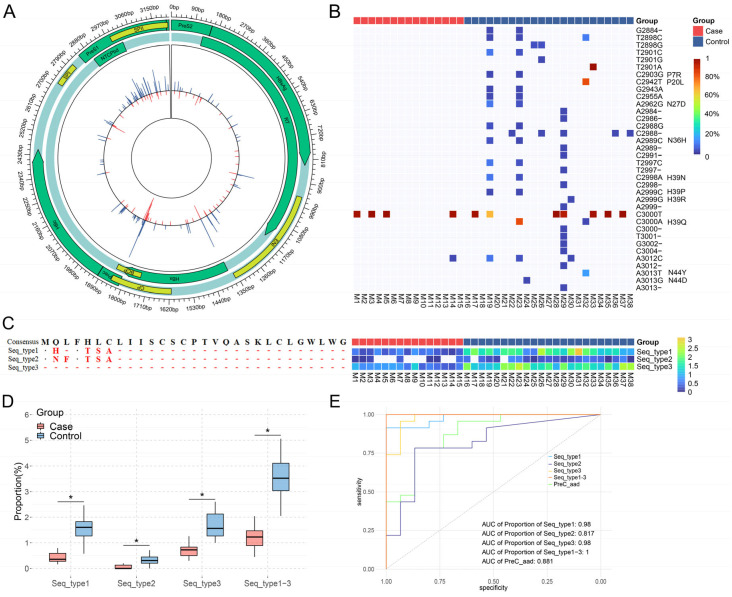
The quasispecies characteristics of NTCPbd and PreC regions between the two groups of mothers. (**A**) The single-position complexity of maternal HBV genome between the case group (red) and the control group (blue). (**B**) The mutations at 25 significantly different positions in the NTCPbd region between the two groups of mothers. (**C**) Alignment on the seq_types 1–3 sequences of the PreC region and their proportions in HBV quasispecies of each mother. (**D**) Comparison of the proportion of seq_types 1–3 between the two groups of mothers. * Represents *p* < 0.05, Mann–Whitney *U* test. (**E**) The ROC curves for the aad of the maternal HBV PreC region and the proportion of seq_types 1–3 in predicting the risk of infant OBI.

**Figure 4 viruses-16-01104-f004:**
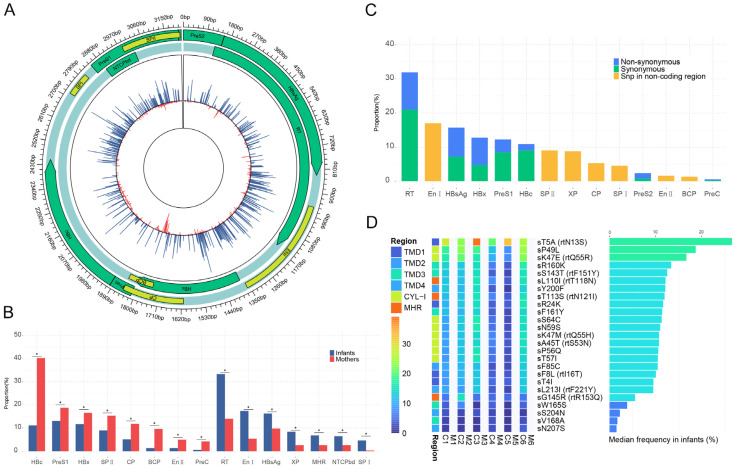
The complexity of single position and the mutations between OBI infants and their mothers. (**A**) The single-position complexity of the HBV genome between infants (blue) and their mothers (red). (**B**) The distribution of positions with significant differences between the two groups. * Represents *p* < 0.05, χ2/Fisher’s exact test. (**C**) The distribution of mutations that were only present in all six infants. (**D**) The distribution and frequency of amino-acid substitutions of HBsAg and the corresponding amino-acid substitutions of RT that were only present in all six infants (ranked by median frequency in infants). TMD, transmembrane domain; CYL-I, the first cytosolic loop.

**Figure 5 viruses-16-01104-f005:**
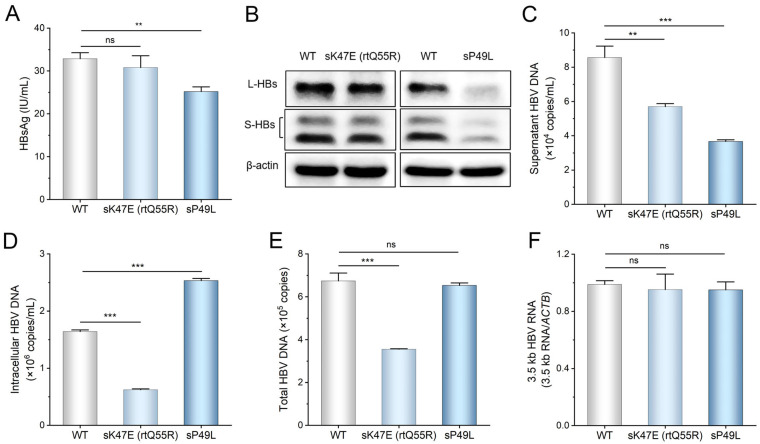
The effect of sK47E (rtQ55R) and sP49L mutations on HBV replication. (**A**) The levels of HBsAg in the supernatants were detected by chemiluminescence immunoassay. (**B**) The intracellular levels of large HBsAg (L-HBs) and small HBsAg (S-HBs) were detected by Western blot. The β-actin protein was used as the internal control. (**C**,**D**) The levels of supernatant and intracellular HBV DNA were detected by qPCR. (**E**) The level of total HBV DNA was the sum of the levels of supernatant and intracellular HBV DNA. (**F**) The levels of intracellular 3.5 kb HBV RNA were detected by RT-qPCR. The *ACTB* mRNA was used as the internal control. “ns” represents “no statistical significance”, ** represents *p* < 0.01, *** represents *p* < 0.001, Student’s *t*-test.

**Table 1 viruses-16-01104-t001:** The baseline characteristics of mothers and infants.

	Case Group	Control Group	*p*
Number (n)	15	23	
Mothers			
Sequence number	861 (843–894.5)	848 (730–993.5)	0.378
Age (years)	23 (20–27)	24 (22–27)	0.141
Alanine aminotransferase (U/L)	16.3 (13.3–23)	18 (15.1–23)	0.226
HBsAg (log_10_IU/mL)	4.56 (4.37–4.73)	4.33 (3.93–4.66)	0.23
HBeAg (log_10_S/CO)	3.15 (3.04–3.19)	3.11 (3.02–3.18)	0.56
HBV DNA (log_10_IU/mL)	8.55 (8.15–8.75)	8.13 (7.79–8.66)	0.21
Infants			
Sex (male/female)	7/8	15/8	0.258
Delivery methods (caesarean/vaginal)	8/7	16/7	0.311
Feeding patterns (breast/artificial)	2/13	2/21	1
Anti-HBs at 7 months (mIU/mL)	936.51 (609.89–1644.29)	800.85 (290.04–1854.97)	0.276
Anti-HBs at 12 months (mIU/mL)	288.91 (172.97–1139.38)	220.13 (63.23–660.08)	0.378
Anti-HBc at 7 months (S/CO)	3.16 (0.71–6.77)	5.72 (1.56–8.14)	0.299
Anti-HBc at 12 months (S/CO)	0.34 (0.14–1.07)	0.2 (0.12–0.63)	0.665

## Data Availability

The raw data of TGS and NGS sequences have been deposited to the National Center for Biotechnology Information (NCBI) (No. PRJNA928585).

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
