# Peer review of "The Characteristic of HBV Quasispecies Is Related to Occult HBV Infection of Infants Born to Highly Viremic Mothers"

_viruses, 2024, doi:10.3390/v16071104_

Round 1

Reviewer 1 Report

Comments and Suggestions for Authors

Yi Li and coworkers present a very interesting study on a considerable series of infants who developed occult HBV infection (OBI) after birth from highly viremic mothers untreated with antivirals. The study dwells on the study of virological factors and risk predictors for OBI in infants and authors report the interesting finding of diversity and complexity of maternal HBV quasispecies that in the case group were lower than those in the control group. Mutations with significant differences between the two groups were most enriched in the NTCPbd and 18 PreC regions and especially the combination of three amino acid mutations in the PreC region associated with OBI in infants. Authors speculate that mutations in the RT and HBsAg regions might contribute to OBI through inhibiting the production of HBV-DNA and HBsAg, respectively. However a very important issue has not been addressed is whether the immune response in the newborns was influenced as well and how by such very interesting differences in quasispecies and mutations. Authors should test the infant sera even sera obtained late on after birth for total anti-HBc antibody and anti-HBs using a quantitatives assay or performing end-point dilution with a qualitative assays to obtain the antibody titers. They should then study  the correlations and/or difference between presence and levels of anti-HBs/anti-HBc in both cases and controls  infants.  In addition it is of paramount importance to know the outcome of HBV infection in terms of serum positivity or negativity of all the 4 major HBV markers (HBsAg, anti-HBs, anti-HBc and HBV-DNA) at 6, 12 months or longer after birth in cases and controls. 

Comments on the Quality of English Language

Good quality

Author Response

Yi Li and coworkers present a very interesting study on a considerable series of infants who developed occult HBV infection (OBI) after birth from highly viremic mothers untreated with antivirals. The study dwells on the study of virological factors and risk predictors for OBI in infants and authors report the interesting finding of diversity and complexity of maternal HBV quasispecies that in the case group were lower than those in the control group. Mutations with significant differences between the two groups were most enriched in the NTCPbd and PreC regions and especially the combination of three amino acid mutations in the PreC region associated with OBI in infants.

Comments 1: Authors speculate that mutations in the RT and HBsAg regions might contribute to OBI through inhibiting the production of HBV-DNA and HBsAg, respectively. However, a very important issue has not been addressed is whether the immune response in the newborns was influenced as well and how by such very interesting differences in quasispecies and mutations.

Response 1: Thanks for your thoughtful and insightful suggestions. According to the baseline characteristics of our cohort (Table 1), the anti-HBs titers at 7 and 12 months of age were comparable between the case and control infants, although the diversity and complexity of maternal HBV quasispecies in the case group were lower than those in the control group. Therefore, we think that the specific immune response is not influenced by the differences in quasispecies and mutations. The corresponding content has been added in the revised manuscript (Lines 367-370).

Comments 2: Authors should test the infant sera even sera obtained late on after birth for total anti-HBc antibody and anti-HBs using a quantitative assay or performing end-point dilution with a qualitative assay to obtain the antibody titers. They should then study the correlations and/or difference between presence and levels of anti-HBs/anti-HBc in both cases and controls infants.

Response 2: Thanks for the reviewer’s thoughtful suggestion. In this study, the blood samples of infants at 7 and 12 months of age were collected, but those of infants at birth, or <1 month were not collected due to the ethical review. The serum quantitative anti-HBs and semi-quantitative anti-HBc in infants at 7 and 12 months of age were tested using Abbott i2000 system. As shown in Table 1, no differences were found in anti-HBs titers at 7 months [936.51 (609.89-1644.29) mIU/mL vs. 800.80 (290.04-1854.97) mIU/mL, P=0.276] and 12 months [288.91 (172.97-1139.38) mIU/mL vs. 220.13 (63.23-660.08) mIU/mL, P=0.378] between the case and control infants. There were also no differences found in anti-HBc at 7 months [3.16 (0.71-6.77) vs. 5.72 (1.56-8.14), P=0.299] and 12 months [0.34 (0.14-1.07) vs. 0.2 (0.12-0.63), P=0.665] between the case and control infants. Therefore, there is no correlation between the presence and levels of anti-HBs/anti-HBc in both the case and control infants. The corresponding content has been added in the revised manuscript (Table 1 and Lines 154-155).

Comments 3: In addition, it is of paramount importance to know the outcome of HBV infection in terms of serum positivity or negativity of all the 4 major HBV markers (HBsAg, anti-HBs, anti-HBc and HBV-DNA) at 6, 12 months or longer after birth in cases and controls.

Response 3: Thanks for the reviewer’s thoughtful suggestion. This study mainly explored the influence of maternal HBV quasispecies on infant OBI. The long-term observation of OBI infants was explored by our previous studies which followed up from 7 months to 3 and 8 years of age (Lu Y, et al. PLoS One 2016, 11:e0166317; Li Y, et al. J Clin Transl Hepatol 2023, 11:661-669). In these studies, OBI was characterized by intermittent viremia. It has been reported that individuals with OBI may experience HBV reactivation in an immunosuppressed state and even develop cirrhosis and hepatocellular carcinoma under the procancer mechanism of OBI (Raimondo G, et al. J Hepatol 2019, 71:397-408; Mak LY, et al. J Hepatol 2020, 73:952-964; Liu C, et al. BMC Med 2022, 20:279; Alameel T, et al. Clin Gastroenterol Hepatol 2021, 19:621–622). Therefore, OBI may pose a significant threat to the prevention and management of HBV infection, which needs to be controlled (Lines 345-352).

Reviewer 2 Report

Comments and Suggestions for Authors

The study used the most advanced knowledge and technologies of molecular biology for an in-depth and detailed analysis of HBV quasispecies in the context of the unsolved problem of neonatal transmission of the virus. The results may have useful and important practical applications in the prevention of maternal-fetal transmission of HBV. In addition to clarifying the mechanisms of transmission itself. As a clinician, I appreciated this manuscript for its scientific rigor and the clarity with which it is presented even to those who are not experts in these sophisticated laboratory investigation techniques. I have no further suggestions for improving the manuscript, which in my opinion is already excellent.

Author Response

Comments: The study used the most advanced knowledge and technologies of molecular biology for an in-depth and detailed analysis of HBV quasispecies in the context of the unsolved problem of neonatal transmission of the virus. The results may have useful and important practical applications in the prevention of maternal-fetal transmission of HBV. In addition to clarifying the mechanisms of transmission itself. As a clinician, I appreciated this manuscript for its scientific rigor and the clarity with which it is presented even to those who are not experts in these sophisticated laboratory investigation techniques. I have no further suggestions for improving the manuscript, which in my opinion is already excellent.

Response: Thanks a lot for the reviewer’s encouragement and enjoy of our work.